# On Optimal Imaging Angles in Multi-Angle Ocean Sun Glitter Remote-Sensing Platforms to Observe Sea Surface Roughness

**DOI:** 10.3390/s19102268

**Published:** 2019-05-16

**Authors:** Dazhuang Wang, Liaoying Zhao, Huaguo Zhang, Juan Wang, Xiulin Lou, Peng Chen, Kaiguo Fan, Aiqin Shi, Dongling Li

**Affiliations:** 1School of Computer Science and Technology, Hangzhou Dianzi University, Hangzhou 310018, China; 13777848996@163.com (D.W.); zhaoly@hdu.edu.cn (L.Z.); 2State Key Laboratory of Satellite Ocean Environment Dynamics, Second Institute of Oceanography, Ministry of Natural Resources, Hangzhou 310012, China; wangjuan@sio.org.cn (J.W.); lxl@sio.org.cn (X.L.); chenpeng@sio.org.cn (P.C.); van.fkg@tom.com (K.F.); aiqinshi@sio.org.cn (A.S.); ldl@sio.org.cn (D.L.); 3College of Oceanography, Hohai University, Nanjing 210098, China; 4Ocean College, Zhejiang University, Zhoushan 316021, China

**Keywords:** sun glitter, sea surface roughness, multi-angle remote-sensing platform, imaging geometry, optimal imaging angle

## Abstract

Sea surface roughness (SSR) is a key physical parameter in studies of air–sea interactions and the ocean dynamics process. The SSR quantitative inversion model based on multi-angle sun glitter (SG) images has been proposed recently, which will significantly promote SSR observations through multi-angle remote-sensing platforms. However, due to the sensitivity of the sensor view angle (SVA) to SG, it is necessary to determine the optimal imaging angle and their combinations. In this study, considering the design optimization of imaging geometry for multi-angle remote-sensing platforms, we have developed an error transfer simulation model based on the multi-angle SG remote-sensing radiation transmission and SSR estimation models. We simulate SSR estimation errors at different imaging geometry combinations to evaluate the optimal observation geometry combination. The results show that increased SSR inversion accuracy can be obtained with SVA combinations of 0° and 20° for nadir- and backward-looking SVA compared with current combinations of 0° and 27.6°. We found that SSR inversion prediction error using the proposed model and actual SSR inversion error from field buoy data are correlated. These results can provide support for the design optimization of imaging geometry for multi-angle ocean remote-sensing platforms.

## 1. Introduction

In ocean optical remote sensing, at certain geometric imaging angles, the sea–air interface experiences specular reflection to form sun glitter (SG) [1]. SG is direct solar reflection from the sea surface and is considered the cause of serious data loss, which can be removed using certain methods in ocean remote sensing [2,3,4]. However, multi-angle SG can significantly promote the observation of sea surface roughness (SSR) through multi-angle remote-sensing platforms and SSR is a key physical parameter in studies on air–sea interactions and ocean dynamics processes. Therefore, when observing ocean dynamic processes, studies must consider the design optimization of imaging geometry for multi-angle ocean remote-sensing platforms. Cox and Munk [1] performed field experiments and developed a model (CM model) to manifest the mathematical relationship between SG radiance and the SSR-related wind-generated mean square of slope using a symmetric slope probability density function, as well as the relationship between SSR and wind speed. Gordon [5] proposed an SG radiation transmission model, and reported that SG intensity depends on the observation geometry and the probability distribution of the slopes of the reflecting facets on the ocean surface. Since then, many scholars have been paying attention to the observation and research of SG. With the deepening of SG remote-sensing research, scientists have found that the intensity of SG is very sensitive to sensor viewing angles (SVA), and have pointed out that ocean internal waves [6,7,8], oil spill pollution [7,8,9], and underwater topography [10,11] exhibit the phenomenon of light and dark reversal of streaks or plaques on SG images. Jackson and Alpers [8] studied and determined the existence of the inversion critical angle. Lu et al. [9] further analyzed the effects of different refractive indices of oil traces and atmospheric effects on the calculation of the critical angle.

Through single-angle SG remote sensing, scholars have noticed the importance of multi-angle SG remote sensing. Matthews [6] used the Advanced Spaceborne Thermal Emission and Reflection Radiometer (ASTER) sensor stereo images to study internal waves, swell waves, bottom topography, and suspended sediment transport in nadir- and backward-looking views. Bréon and Henriot [12] retrieved the wind from multi-angle (14 angles) images from the Polarization and Directionality of the Earth’s Reflectance (POLDER) instrument. Similarly, the Multi-angle Imaging SpectroRadiometer (MISR) was found to be capable of discriminating oil spills and improving the operational monitoring of oil releases [13]. Fox et al. [14] explored the relationship between the MISR-observed width of the sun glint pattern over ocean waters and the near-surface wind speed, and the acquired data were then used to develop an algorithm for wind retrieval. Matthews et al. [15] conducted a detailed analysis of brightness reversal in the SG signature of a lake bed topographic feature observed within stereo ASTER data. Harmel and Chami [16] developed a scheme to retrieve wind speed directly from a passive satellite sensor in the visible/near infrared bands by an iteration process. Yang et al. [11] and Zhang et al. [17] studied the shallow sea sand wave topography based on multi-angle SG, and showed the difference in SG radiance along the scan line with different SG imaging geometries. They confirmed that SG is very sensitive to SVA. Kudryavtsev et al. [18,19] developed a practical method for retrieving directional spectra of ocean surface waves quantitatively from near-multidirectional SG imagery from the staggered detectors of the Copernicus Sentinel-2 Multi-Spectral Instrument. Zhang et al. [20] constructed a multi-angle SG remote-sensing model using ASTER stereo multi-angle SG images, and established an SSR estimation model.

To take better advantage of multi-angle SG remote sensing, multi-angle ocean sun glitter remote-sensing platforms can be designed concurrently with the development of unmanned aerial vehicle formations and satellite constellations [21]. However, multi-angle SG remote-sensing detection applications and SSR quantitative inversion studies can only use satellite remote-sensing platforms with fixed SVA combinations. Due to the sensitivity of the SG to SVA, studies must extensively explore the selection of the correct optimal imaging angle. Therefore, considering the SG radiation error propagation, this study constructed an error transfer simulation model based on the multi-angle SG remote-sensing radiation transmission and SSR estimation models and implemented the SSR inversion errors at different imaging geometry combinations to evaluate the optimal imaging angle combination. Finally, we discuss the design optimization potential that the imaging geometry has on the multi-angle remote-sensing platforms for satellite constellations and drone platforms.

## 2. Materials and Methods

### 2.1. Estimation Model for Sea Surface Roughness (SSR)

For ASTER two-angle images, Zhang et al. [20] studied and established an SSR estimation model based on multi-angle SG using ASTER channel 3N and channel 3B images. The model was determined as follows:(1)$$\sigma_{0}^{2}=\frac{\tan^{2}\beta_{B}-\tan^{2}\beta_{N}}{\mathrm{Ln}\left(\frac{L_{gN}}{L_{gB}}\;\frac{R\left(\omega_{B}\right)\cos\theta_{N}\cos^{4}\beta_{N}}{R\left(\omega_{N}\right)\cos\theta_{B}\cos^{4}\beta_{B}}\right)}$$where $\sigma_{0}^{2}$ is SSR, θ is sensor viewing angle (SVA) ($\theta_{N}$ in 3N image and $\theta_{B}$ in 3B image), ϕ is sensor azimuth angle ($\phi_{N}$ in 3N image and $\phi_{B}$ in 3B image), $\theta_{0}$ is sun zenith angle, and $\phi_{0}$ is sun azimuth angle.

R(ω) denotes the Fresnel reflection coefficient ($R\left(\omega_{N}\right)$ in 3N image and $R\left(\omega_{B}\right)$ in 3B image), and it was determined as follows:(2)$$R\left(\omega \right)=\frac{1}{2}\left[\frac{\sin^{2}\left(\omega -\omega {}^{\prime }\right)}{\sin^{2}\left(\omega +\omega {}^{\prime }\right)}+\frac{\tan^{2}\left(\omega -\omega {}^{\prime }\right)}{\tan^{2}\left(\omega +\omega {}^{\prime }\right)}\right]$$where ω and $\omega {}^{\prime }$ are the incident angle and refraction angle of the sun’s rays on the inclined surface of the sea. The relationship between them is as follows:(3)$$\cos\left(2\omega \right)=\cos\theta \cos\theta_{0}+\sin\theta \sin\theta_{0}\cos\left(\phi -\phi_{0}\right)$$
(4)$$\sin\omega {}^{\prime }=\frac{\sin\omega }{1.34}$$

Here, let Δϕ be the relative azimuth of the sensor and the sun; it is defined as:(5)$$\Delta \phi =\phi -\phi_{0}$$

β denotes the surface tilt angle of a facet on the sea surface ($\beta_{N}$ in 3N image and $\beta_{B}$ in 3B image), it is defined as:(6)$$\cos\beta =\frac{\cos\theta +\cos\theta_{0}}{2\cos\omega }$$

$L_{g}$ is SG radiance ($L_{gN}$ in 3N image and $L_{gB}$ in 3B image) according to Gordon [5]:(7)$$L_{g}=F_{0}T\frac{R\left(\omega \right)}{4\cos\theta \cos^{4}\beta }P\left(Z_{x},Z_{y}\right)$$where T denotes the downwelling direct transmittance, according to Bréon and Henrio [12]. In the open sea, the value of T is about 0.85. $F_{0}$ denotes solar irradiance at the top of the atmosphere according to Neckel [22]. It can be calculated as follows:(8)$$F_{0}\left(sday\right)={\bar{F}}_{0}\left[1+0.0167\cos\left(\frac{2\pi \left(sday-3\right)}{365}\right)\right]$$where sday is Julian Day, and ${\bar{F}}_{0}$ is the average of many years of $F_{0}$. According to the latest recommendation of the International Bureau of Weights and Measures, the value is $1367\;W/m^{2}$.

Normalized SG radiance (I) is defined as follows:(9)$$I=\frac{L_{g}}{F_{0}T}=\frac{R\left(\omega \right)}{4\cos\theta \cos^{4}\beta }P\left(Z_{x},Z_{y}\right)$$where $P\left(Z_{x},Z_{y}\right)$ is the probability density function of the wave slope as functions of individual slope components $Z_{x}$ and $Z_{y}$. It is determined as follows:(10)$$P\left(Z_{x},Z_{y}\right)=P\left(\beta \right)=\frac{1}{\pi \left(\sigma_{0}^{2}\right)}\exp\left(-\frac{\tan^{2}\beta }{\sigma_{0}^{2}}\right)$$

According to Shao et al. [23], when the wind speed is lower than 15 m/s, the relationship between SSR and sea surface wind speed is as follows:(11)$$\sigma_{0}^{2}=0.003+0.00512W$$

The SVA of the ASTER 3N and 3B images selected by Zhang et al. [20] were fixed values ($\theta_{N}$ = 0° and $\theta_{B}$ = 27.6°), and the purpose of this study was to verify whether an optimal SVA combination that has the best effect on the SSR estimation model of multi-angle SG remote sensing could be determined. The SSR estimation model was modified to the following general situation:(12)$$\sigma_{0}^{2}=\frac{\tan^{2}\beta_{II}-\tan^{2}\beta_{I}}{\mathrm{Ln}\left(\frac{L_{gI}}{L_{gII}}\;\frac{R\left(\omega_{II}\right)\cos\theta_{I}\cos^{4}\beta_{I}}{R\left(\omega_{I}\right)\cos\theta_{II}\cos^{4}\beta_{II}}\right)}$$

I and II respectively represent the remote-sensing image related information under two different SVA.

### 2.2. Sun Glitter (SG) Geometry in Stereo Images

The sun zenith angle and sun azimuth can be calculated by longitude, latitude, and time. Because multiple imaging times of multi-angle SG remote sensing are close, we believe that the angles associated with the sun have not changed, but SVA and sensor azimuth are significantly different. Referring to Matthews’ method [6], Yang et al. [11] and Zhang et al. [20] developed a method for calculating remote-sensing geometric parameters of pixel-by-pixel elements on both sides of the nadir. On this basis, we set the SVA positive and negative. Figure 1 shows the geometric model of multi-angle SG imaging.

Regarding the calculation of geometric angles, the left and right sides of the nadir should be considered. In the calculation of SVA and the sensor zenith angle of any pixel, suppose each pixel between this pixel and the center of the scan line (where the pixel is located) is numbered *n*, then *n* is positive when this pixel is at the left of the center and negative on the right side.

In order to facilitate the description and calculation of geometric parameters, positive and negative SVA are added. With the sensor in the flight direction, the SVA is positive when the remote-sensing image is acquired forward, corresponding to the position of c in Figure 1 (forward-looking); Conversely, the SVA is negative when the remote-sensing image is acquired backwards, corresponding to the position of a (backward-looking); b corresponds to the position of the ASTER 3N image (nadir-looking).

The nadir-looking viewing angle ($\theta_{b}$) is given by:(13)$$\theta_{b}=\left|n\times \left(IFOV\right)+P\right|$$where *IFOV* is the instantaneous field of view, and *P* is pointing angle.

The azimuth angle is 0° in the north direction of the target and gradually increases in the clockwise direction within a range of 0°–360°. The sensor azimuth angle of a pixel in the nadir-looking direction is different on different sides of nadir. When the pixel is on the left side of nadir, the nadir-looking azimuth angle ($\phi_{b}$) is given by:(14)$$\phi_{b}=270{}^{\circ}+S$$

When the pixel is on the right side of nadir, $\phi_{b}$ is given by:(15)$$\phi_{b}=90{}^{\circ}+S$$

The geometric parameters of the backward-looking (a) and forward-looking (c) views are calculated differently from the nadir-looking (b) view. The backward-looking viewing angle ($\theta_{a}$) and forward-looking viewing angle ($\theta_{c}$) are given by:(16)$$\theta_{\left(ac\right)}=\tan^{-1}\left(\frac{\sqrt{{\left(h\tan P+mn\right)}^{2}+{\left(h\tan v/\cos P\right)}^{2}}}{h}\right)$$where *h* denotes the satellite height, *m* denotes the image spatial resolution, V denotes the angle between the nadir-looking view and the current view. The positive and negative of V are consistent with the positive and negative of current SVA.

The sensor azimuth angle of a pixel in the backward-looking or forward-looking view is also different on different sides of nadir. When the pixel is on the left side of nadir, we can obtain $\phi_{a}$ and $\phi_{c}$ through,
(17)$$\phi_{\left(ac\right)}=270{}^{\circ}-\tan^{-1}\left(\frac{-1\times \tan v}{\tan\left(n\times \left(IFOV\right)+P\right)}\right)+S$$

When the pixel is on the right side of nadir, $\phi_{a}$ and $\phi_{c}$ are given by:(18)$$\phi_{\left(ac\right)}=90{}^{\circ}-\tan^{-1}\left(\frac{-1\times \tan v}{\tan\left(n\times \left(IFOV\right)+P\right)}\right)+S$$

### 2.3. Simulation Model

When using the SSR estimation model based on multi-angle SG, the remote-sensing image information under two SVA is first obtained, and then SSR is estimated by the model. Without considering inherent errors of the sensor and the error of radiation correction, accurate SG radiance of different observation geometries can always be obtained, thus providing accurate SSR information. In fact, the above two kinds of errors always exist, and thus different imaging geometry combinations involve different transmission processes for errors. We establish an error transfer simulation model, as shown in Figure 2. We use python programming language to design simulation software program based on the above models, and analyze the error transmission situations under different imaging geometry combinations to evaluate the optimal observation geometry combination by the program.

The process is explained as follows:
①Input model parameters and determine sensor pointing angle (*P*), scene orientation angle (*S*), sun zenith angle ($\theta_{0}$), sun azimuth angle ($\phi_{0}$), and SVA combination ($\theta_{I}$,$\theta_{II}$);②Input the sea surface wind speed (*W*), use the CM model [1] to calculate SSR attributable to sea surface wind ($\sigma_{0}^{2}$), and use the SSR as the real value of the assessment;③Combine the sensor information input in ①, calculate sensor azimuth angles ($\phi_{I}$,$\phi_{II}$) under SVA combinations ($\theta_{I}$,$\theta_{II}$) according to Equations (14), (15), (17) and (18);④Simulate a pair of normalized SG radiance ($I{}^{\prime }$, $I{}^{\prime \prime }$) (Equation (9));⑤Add the simulated error (ΔI) to the normalized SG radiance ($I{}^{\prime }$, $I{}^{\prime \prime }$). Here, we use a 5% multiplicative error and an additive error of 0.00005 according to the signal-to-noise ratio of the ocean optical remote sensor and the general radiation correction error (ΔI=I*5%+0.0005), and get a pair of normalized SG radiance with errors ($f\left(I{}^{\prime }\right)$, $f\left(I{}^{\prime \prime }\right)$).⑥Apply $f\left(I{}^{\prime }\right)$, $f\left(I{}^{\prime \prime }\right)$ and related parameters to the SSR estimation model based on multi-angle SG (Equation (12)). Then get the estimated SSR ($(\sigma_{0}^{2}$)’);⑦Calculate the error ($\Delta \sigma_{0}^{2}$) between the estimated SSR ($(\sigma_{0}^{2}$)’) and the real SSR ($\sigma_{0}^{2}$) attributable to wind speed.

The result obtained ($\Delta \sigma_{0}^{2}$) using the above model is the result of the SG radiance error transmitted in the SSR estimation model. In this paper, the expression of $\Delta \sigma_{0}^{2}$ uses the percentage error of $(\sigma_{0}^{2}$)’ and $\sigma_{0}^{2}$. The smaller $\Delta \sigma_{0}^{2}$, the better the SVA combination under such environmental factors (sun zenith angle, sun azimuth angle, wind speed). Conversely, if the value of $\Delta \sigma_{0}^{2}$ is large, this combination should be avoided.

## 3. Simulation and Analysis

The formation of SG is closely related to some imaging parameters, and studies have shown that SG is very sensitive to SVA [9,10]. In this study, the ASTER data are used as the basic reference. By adjusting some key parameters, different sensor parameters and different environmental parameters were simulated. The influence of SVA in the range (−90°, 90°) and in steps of 0.5° on the SG radiation transmission was analyzed, and effect of SVA combinations on inversion results of SSR was investigated by simulating the distribution of the relative error ($\Delta \sigma_{0}^{2}$) under different SVA combinations. According to the simulation results under most conditions, in the distribution of error transfer, only SVA in the range [−50°, 50°] were displayed. In actual remote-sensing images, in order to ensure the accuracy of SSR estimation, the SG radiation should be at least not less than the sum of Rayleigh scattering and aerosol scattering. According to Yang et al. [11], the normalized SG radiance threshold is 0.004. When the normalized SG radiance exceeds this threshold, it can be used for SSR estimation. In order to analyze the applicability of SVA combinations under different conditions, we separately counted the number of SVA combinations $n_{10}$ and $n_{20}$ under the conditions of $\Delta \sigma_{0}^{2}$ < 10% and $\Delta \sigma_{0}^{2}$ < 20%. The total number n of SVA combinations in the simulation range was: (180×2−1)×(180×2−1)=128,881. Then, the applicable probabilities of SVA combinations, named as $\rho_{10}$ and $\rho_{20}$, were obtained by calculating the ratios between $n_{10}$ and n, and $n_{20}$ and n. With higher probability, the SSR estimation model is more effective, and thus, model parameters should be adjusted to increase this possibility. The general settings of the parameters used in the simulation are shown in Table 1. In order to explore the influence of imaging geometric parameters and different environments on SSR inversion results, we mainly simulated changes in four key parameters: pointing angle (*P*), sea surface wind speed (*W*), sun azimuth angle ($\phi_{0}$), and sun zenith angle ($\theta_{0}$). Then, we analyzed the optimal SVA combinations under different parameter changes.

### 3.1. Pointing Angle

Due to the pointing angle (*P*), sensor azimuth is changed, and the nadir is offset. According to the characteristics of ASTER, the range of *P* is −24°≤P≤24°. When *P* is positive, the sensor swings to the left in the flight direction, and when it is negative, the sensor swings to the right in the flight direction. When *P* is in the range of [−24°, 24°] and in steps of 6°, the trend of normalized SG radiance at different SVA is simulated, as shown in Figure 3.

In the imaging parameters, the sun azimuth is 90°, which is the position of the sun in the east. According Equations (14), (15), (17) and (18), when *P* is 24°, the sensor azimuth is about 270°, that is, the sensor takes a remote-sensing image against the sunlight, and thus SG radiance is strong at this time. Conversely, when *P* is −24°, the sensor takes a remote-sensing image along the sunlight, and thus SG radiance is extremely weak. Considering the SG radiance threshold of 0.004, the range of SVA that can be used for the SSR estimation model was found to increase with higher *P*. When *P* = 24°, the range of SVA that can be used is about −35°≤θ≤30°. When P≤−12°, the SG radiance is completely below the threshold, and the range of SVA that can be used is unavailable.

Under different *P* values, the distribution of $\Delta \sigma_{0}^{2}$ at different SVA combinations is shown in Figure 4, and the statistical results of $\rho_{10}$ and $\rho_{20}$ are shown in Figure 5. According to Figure 4, when *P* is negative, the overall SSR estimation model is ineffective, and at an appropriate value, some optimal combinations of SVA are always available. According to Figure 5, when P≤−12°, both $\rho_{10}$ and $\rho_{20}$ approach 0. When *P* gradually increases, the values of $\rho_{10}$ and $\rho_{20}$ also increase gradually, and the rate of increase gradually weakens with $\rho_{20}$ being always about 1.5% higher than $\rho_{10}$. When *P* = 24°, $\rho_{10}$ is about 9.7% and $\rho_{20}$ is about 11.0%.

### 3.2. Wind Speed

Ocean dynamics processes are the cause of SSR, among which sea surface wind is the most important process. The sea surface wind field is an important physical parameter for the interaction between the ocean and the atmosphere, and it is the main driving force of upper seawater movement. The trend of normalized SG radiance at different SVA is simulated considering sea surface wind speed (*W*) in the range of [1 m/s, 15 m/s] and in steps of 2 m/s, as shown in Figure 6.

At high sea surface wind speeds, SSR is high. The simulation results show that the overall SG radiance is also stronger at high wind speeds. With the SG radiance threshold (0.004), the range of SVA that can be used for the SSR estimation model was found to increase with increasing wind speed. At *W* = 15 m/s, the range of available SVA is about −41°≤θ≤36°, and at *W* = 1 m/s, the range of available SVA is very narrow at −7°≤θ≤2°.

At different wind speeds, the distribution of $\Delta \sigma_{0}^{2}$ under different SVA combinations is shown in Figure 7, and the statistical results of $\rho_{10}$ and $\rho_{20}$ are shown in Figure 8. According to Figure 7, the SSR estimation model becomes more effective as the wind speed increases. In addition, an optimal combination of SVA could be always selected at all wind speeds. According to Figure 8, when *W* = 1 m/s, $\rho_{10}$ and $\rho_{20}$ are close to 0. When *W* gradually increases, the values of $\rho_{10}$ and $\rho_{20}$ increase gradually, but the increase trend of $\rho_{20}$ is greater than that of $\rho_{10}$. At *W* = 15 m/s, $\rho_{10}$ is about 12.9% and $\rho_{20}$ is about 16.2%. This shows that, the number of available SVA combinations is small at low wind speeds, but it is large at high wind speeds.

### 3.3. Sun Azimuth Angle

Sun azimuth ($\phi_{0}$) is another important imaging geometric parameter. It varies greatly at different times of the day, different seasons of the year, and different geographical locations, such as in the Southern Hemisphere and the Northern Hemisphere. In the actual environment, it is $0{}^{\circ}\leq \phi_{0}<360{}^{\circ}$, and $\phi_{0}$ will also significantly impact SG. In the above simulation process, we set $\phi_{0}=90{}^{\circ}$. Considering $\phi_{0}$ in the range of [0°, 360°) and in steps of 30°, the trend of normalized SG radiance at different SVA is simulated, as shown in Figure 9.

According to Figure 9, the simulation results show that changes in $\phi_{0}$ significantly impact SG radiance. Taking the peak point *P*1 as an example, when the sun azimuth of *P*1 is 0°, the sun is in the north direction. According Equation(14), (15), (17) and (18), when SVA is negative, the sensor azimuth is about 180°, the sensor takes a remote-sensing image against the sunlight, and a strong SG radiance is obtained at this time. When SVA is positive, the sensor azimuth is about 0°, and the sensor takes a remote-sensing image along the sunlight, resulting in weaker SG radiance. At *P*1, SVA is −20° and sun zenith angle is 20°. *P*1 is the complete specular reflection point [8] when the sensor takes the image against the sun. Therefore, *P*1 reaches the peak of SG radiance. Similarly, when the sun azimuth of *P*2 is 180°, the sun is in the south direction. When SVA is positive, the sensor azimuth is about 0°, and *P*2 is the complete specular reflection point when θ=20°. Therefore, *P*2 also reaches the peak of SG radiance. At other sun azimuths, complete specular reflection points do not occur under the ASTER flight orbit. However, under certain SVA, a peak of SG radiance always occurs, and as the sun azimuth changes, the peak increases or decreases according to a certain law. Considering the SG radiance threshold of 0.004, the results show that changes in the sun azimuth significantly affect the range of available SVA for the SSR estimation model. At $\phi_{0}=0{}^{\circ}$, the range of available SVA is about −55°≤θ≤11°, and at $\phi_{0}=180{}^{\circ}$, the range is −11°≤θ≤55°.

Figure 10 shows the distribution of $\Delta \sigma_{0}^{2}$ at different sun azimuth angles under different SVA combinations is shown in, and Figure 11 shows the statistical results of $\rho_{10}$ and $\rho_{20}$. According to Figure 10, the sun azimuth is also a key factor in the selection of the optimal SVA combination. The simulation results show that changes in the sun azimuth not only change the size but also the position of the optimal SVA combination range. According to Figure 11, at $\phi_{0}=0{}^{\circ}$ or $\phi_{0}=180{}^{\circ}$, $\rho_{10}$ is about 11.0% and $\rho_{20}$ is about 12.4%. As the sun moves towards the east or west, $\rho_{10}$ and $\rho_{20}$ gradually decrease, and $\rho_{20}$ is always about 1.5% higher than $\rho_{10}$. At $\phi_{0}=90{}^{\circ}$ or $\phi_{0}=270{}^{\circ}$, $\rho_{10}$ is about 4.9% and $\rho_{20}$ is about 6.1%. In Figure 11, the main distributions of valley peak positions A and C correspond to the southern hemisphere region at noon in winter and the northern hemisphere region at noon in winter. The main distributions of the bottom positions B and D correspond to the mid-latitude area in the morning and afternoon, respectively. The distributions show that the SSR estimation model provides different results at different times and geographical locations.

### 3.4. Sun Zenith Angle

The sun zenith angle ($\theta_{0}$) is another important imaging geometric parameter, and it is the key parameter affecting SG radiance. Zhang et al. [20] used 170 ASTER images to verify an SSR estimation model based on multi-angle SG remote sensing. The screening rules for the selected images are: (1) the season is summer, the local time is around 10:00, (2) the spatial position is at low latitude. The purpose of these screening factors is to obtain better SG images under the appropriate sun zenith angle. Considering $\theta_{0}$ in the range of [10°, 40°] and in steps of 5°, the trend of normalized SG radiance at different SVA is simulated as shown in Figure 12.

According to Figure 12, the simulation results show that SG radiance is strong at small sun zenith angles. This implies that strong SG radiance will be observed in images captured during midday in the summer. Considering the SG radiance threshold of 0.004, the range of available SVA for the SSR estimation model was found to decrease with increasing sun zenith angle. At $\theta_{0}=10{}^{\circ}$, the range of available SVA is about −32°≤θ≤30°, and at $\theta_{0}\geq 35{}^{\circ}$, SG radiance is below the threshold and the range of usable SVA is unavailable.

Figure 13 shows the distribution of $\Delta \sigma_{0}^{2}$ at different sun zenith angles under different SVA combinations, and Figure 14 shows the statistical results of $\rho_{10}$ and $\rho_{20}$. According to Figure 13, the optimal SVA combination is also greatly affected by sun zenith angle. At low sun zenith angles, the SSR estimation model is more efficient. According to Figure 14, $\rho_{10}$ is about 8.7% and $\rho_{20}$ is about 10.0% at $\theta_{0}=10{}^{\circ}$. When the sun zenith angle gradually increases, $\rho_{10}$ and $\rho_{20}$ gradually decrease, reaching close to 0 at $\;\theta_{0}\geq 35{}^{\circ}$.

## 4. Discussion

### 4.1. Comparison of Predicted and Actual Error

To evaluate the validity of the error assessment in this study, we estimated the SSR inversion error based on the actual geometric parameters for the ASTER multi-angle SG remote-sensing images obtained using our model and compared them with the actual SSR inversion error reported in Zhang et al. [20].

Zhang et al. [20] applied ASTER remote-sensing imagery to the SSR estimation model based on multi-angle SG and then inverted the sea surface wind speed based on the estimated SSR. Finally, certain buoy wind speed data from the National Data Buoy Center (NDBC) and inversion wind speed data were selected for comparative analysis. The selected buoy and center positions of the remote-sensing image were within 30 km. The time difference of the buoy data within half an hour was matched and a total of 6 sets of matching data were obtained. A set of actual errors between the estimated and actual wind speeds was obtained through comparisons.

Using the proposed error analysis simulation model, we were able to predict, in advance, the SSR error estimated for the six ASTER remote-sensing images. The relationship between SSR and wind speed (Equation (11)) can be used to convert the estimated SSR error into sea surface wind speed error. First, according to the characteristics of the ASTER sensor, the error (ΔI) of the simulated SG radiance is input into the simulation model. Currently, we consider that ΔI consists of two parts: the inherent sensor error and the radiation correction error. We set ΔI=I*5%+0.0005 as described in the error transfer simulation model above. The predicted error was compared with the actual generated error, whose comparison result is shown in Figure 15.

The predicted and actual errors were found to have a positive correlation (correlation coefficient = 0.85) but there is a large absolute deviation between them. This large absolute deviation is possibly due to the smaller simulated SG radiance error (ΔI) input into the simulation model compared with the true SG radiance error, which results in a small prediction error. On the other hand, Zhang et al. [20] only used a simple atmospheric correction to process the ASTER images. They considered effects from Rayleigh (Lr) and aerosol scattering (La) but ignored both whitecap reflection (Lwc) and water-leaving radiance (Lw). Rayleigh scattering correction was performed using the single scattering model developed by Gordon [24] and both the aerosol optical thickness and scattering phase functions in the aerosol scatter correction were determined using empirical parameter methods. Based on Gordon et al. [24], Zhang et al. [20] considered that their model had only a 10% error for the calculation of a single Rayleigh scattering. Atmospheric correction of ASTER remote-sensing images is a complex but non-negligible process. Oversimplified atmospheric corrections may lead to large errors in the inversion of sea surface and actual wind speeds. Therefore, the actual error is much larger than the prediction error. To further improve the estimation accuracy of the simulation model, we analyzed ΔI at different conditions.

### 4.2. Optimal Relative Azimuth Angles in the Multi-Angle Remote-Sensing Platform

In the simulation model, the sun azimuth angle and sensor azimuth angle only affect the relative azimuth angle (Δϕ) (Equation (5)). In future SSR estimation model applications, such as drone formation, the platform would allow for more flexibility in adjusting and controlling the sensor parameters compared with satellite platforms that have fixed orbital parameters. Aviation platforms can change their flight path and, thus, for a given sun azimuth angle, they can change the sensor azimuth angle to optimize the relative azimuth angle. To analyze the influence that the sensor azimuth has on the SSR estimation results, we chose different latitudes or simulation. The locations of 5°, 20°, and 40° N at 1030 local time on 21 August 2018 were selected as the background, and the imaging geometry and environment parameters were set as follows:(*P* = 0°, *W* = 5 m/s), and the preset SVA combination is ($\theta_{I}=10{}^{\circ}$, $\theta_{II}=20{}^{\circ}$). Within the sensor azimuth range of [0°, 360°), the distribution of SSR estimation error and normalized SG radiance are simulated, as shown in Figure 16. 

According to the result, it is always possible to reduce $\Delta \sigma_{0}^{2}$ to 10% or less by adjusting the sensor azimuth. And we use the “X” to mark the main optimal area. However, the optimal sensor azimuth combinations changes in different latitude locations, and there are abnormal bands, such as S1 and S2 in Figure 16a. The sensor azimuth combinations marked by “X” can be selected according to actual location and flight detection. When the latitude is higher, the combinations that be marked are less because of the weaker SG radiance, such as location of 40° N. The result shows that optimal sensor azimuth combinations can always obtain strong SG radiance. In the locations of 5°, 20°, and 40° N, respectively, the sun azimuth angles are 71.2°, 105.7° and 137.1°. When SG radiance is strongest, the sensor azimuth is approximately 250°, 285° and 315° respectively. Therefore, the optimal relative azimuth angle (Δϕ) is confirmed to be about 180°. Conversely, when sensor azimuth angles are about 70°, 100° and 130° respectively, $\Delta \sigma_{0}^{2}$ will increase. Without being limited by the sun azimuth, the relative azimuth can be controlled by adjusting the flight direction. Therefore, the flexibility of an aviation platform as a multi-angle remote-sensing platform allows for more optimized designs for imaging angle. According to the normalized SG radiance of the selected profiles, the resulting set of SG radiance are the same under certain sensor azimuth combinations. We mark the corresponding combinations in Figure 16 respectively. The abnormal bands occur when both sensors have the same SG radiances; thus, combinations that cause the same set of SG radiance should be avoided. 

## 5. Conclusions

Through simulation analysis, this study attempted to determine the optimal imaging angle and their combinations for the SSR estimation model, whose results will be used to design multi-angle ocean remote-sensing platforms. The simulation results show that an error occurs in the SSR estimation results with exposure to non-ideal conditions, which is closely related to the SVA combination. Optimal SVA combinations can always be selected but change in different environments. Simulations of the pointing angle (*P*), sea surface wind speed (*W*), and sun zenith angle ($\theta_{0}$) showed that, when the pointing angle is directed towards the sun (*P* > 0°), the wind speed is large and the sun zenith angle is small, there is a generation of stronger SG radiance, and there is an improvement in the ratio ($\rho_{10}$,$\rho_{20}$) of the optimal SVA combination. In such cases, the SSR estimation model has a wider application range for the SVA combination. According to the result of the simulation of four key parameters, in the general environment (i.e., the sun is in the east), high accuracy for the SSR estimation results ($\Delta \sigma_{0}^{2}<20\%$) is possible with SVA combination ($\theta_{I}=0{}^{\circ}$, $\left|\theta_{II}\right|=20{}^{\circ}$) on the ASTER platform.

To verify the error transfer simulation model designed in this study, the prediction error output from the model was compared with the actual error. According to Figure 15, although the errors have a large absolute deviation, they have a positive correlation. Therefore, we were able to verify that the simulation model developed in this study has practical usability. Finally, we discussed the relative azimuth combination in multi-angle remote-sensing platforms. The relative azimuth (Δϕ), as part of the imaging angle, can only be optimally adjusted by the sun azimuth ($\phi_{0}$) on satellite sensors with fixed flight orbits, such as ASTER. The simulation results show that the relative azimuth is optimal as the sun moves towards the south or north. When the sun azimuth is at 0°or 180°, the applicable probabilities of the SVA combinations are highest, where $\rho_{10}$ and $\rho_{20}$ are approximately 11.0% and 12.4%, respectively. Based on Figure 16, we can use the aviation platform as a multi-angle remote-sensing platform and, thus, the sensor azimuth can also be used to optimally adjust the relative azimuth angle to further improve SSR inversion accuracy. 

At present, SSR estimation is one of the important applications in multi-angle SG remote sensing, as well as being a topic of current, in-depth research. In ocean dynamics processes, sea surface winds and shallow sea sand wave topography can cause changes in sea surface roughness and, thus, multi-angle SG remote-sensing images can be used to invert the sea surface wind speed [20] and depth of shallow sea sand wave topography [17]. The use of multi-angle SG can also improve our ability to detect sea surface oil spills [13]. By applying an optimal imaging angle to the design of multi-angle ocean sun glitter remote-sensing platforms, we can improve SSR estimation accuracy. This further demonstrates the application potential of multi-angle SG remote sensing in studies on ocean dynamic processes and the detection of phenomena that cause changes in SSR.

## Figures and Tables

**Figure 1 sensors-19-02268-f001:**
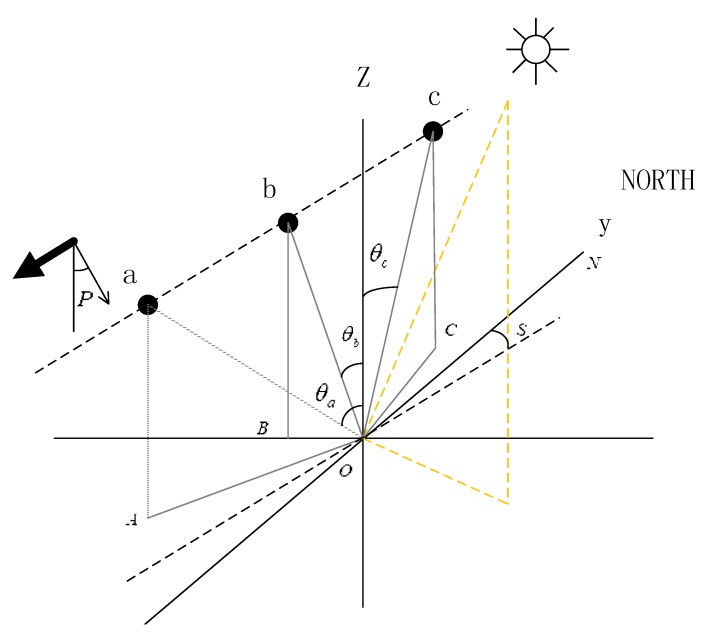
Observational geometry of the Advanced Spaceborne Thermal Emission and Reflection Radiometer (ASTER) stereo images. Sun zenith angle ($\theta_{0}$), sensor nadir-looking viewing angle ($\theta_{b}$), sensor forward-looking viewing angle ($\theta_{c}$), sensor backward-looking viewing angle ($\theta_{a}$), scene orientation angle (*S*), and pointing angle (*P*) is presented.

**Figure 2 sensors-19-02268-f002:**
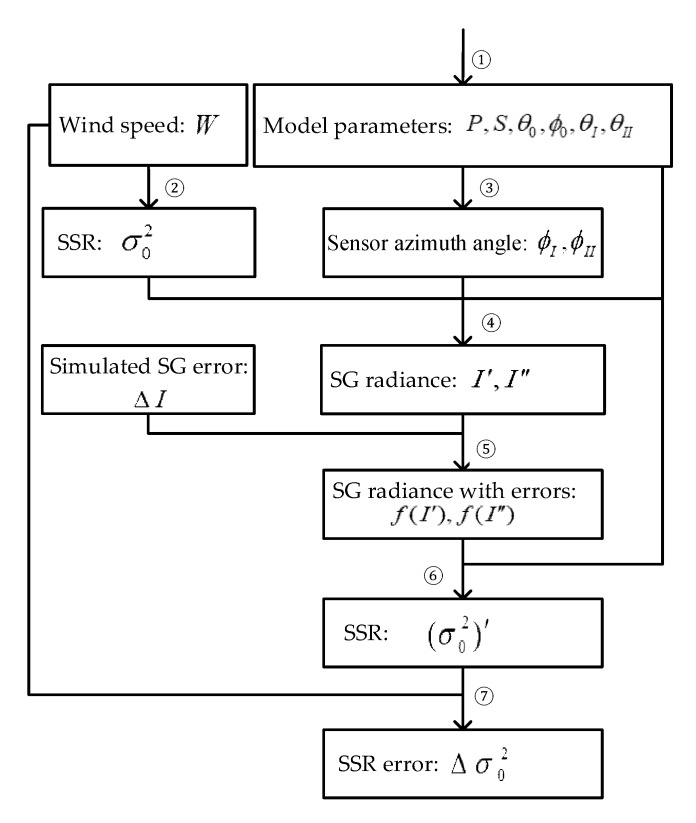
Error transfer simulation model.

**Figure 3 sensors-19-02268-f003:**
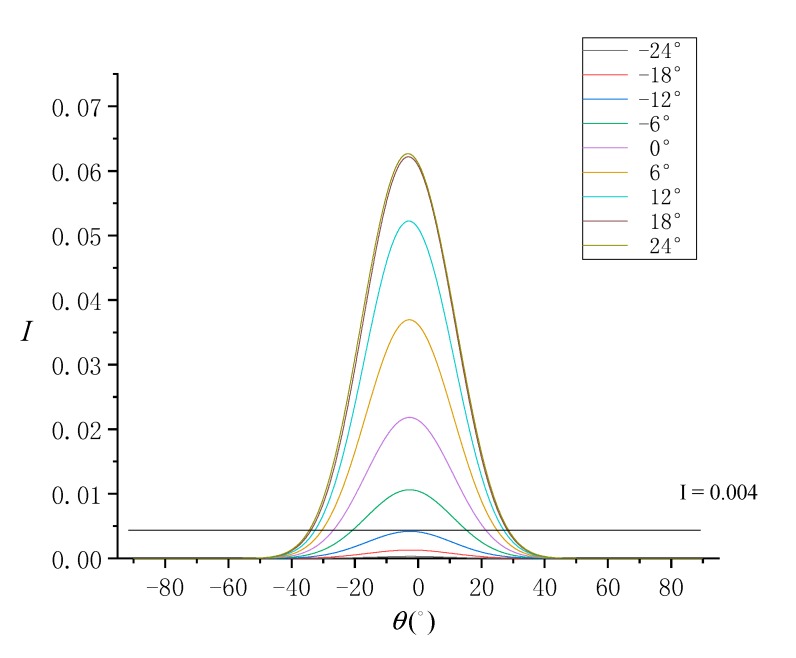
Normalized sun glitter (SG) radiance (I) simulated by each sensor view angle (SVA) (θ) under different pointing angles.

**Figure 4 sensors-19-02268-f004:**
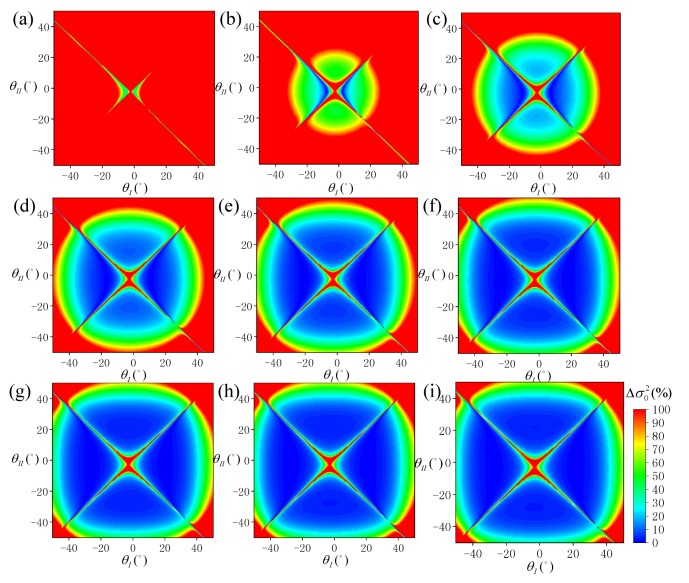
Distribution of sea surface roughness (SSR) estimation error ($\Delta \sigma_{0}^{2}$) simulated under each SVA combination ($\theta_{I}$,$\theta_{II}$) at different pointing angles: (**a**) *P* = −24°, (**b**) *P* = −18°, (**c**) *P* = −12°, (**d**) *P* = −6°, (**e**) *P* = 0°, (**f**) *P* = 6°, (**g**) *P* = 12°, (**h**) *P* = 18°, and (**i**) *P* = 24°.

**Figure 5 sensors-19-02268-f005:**
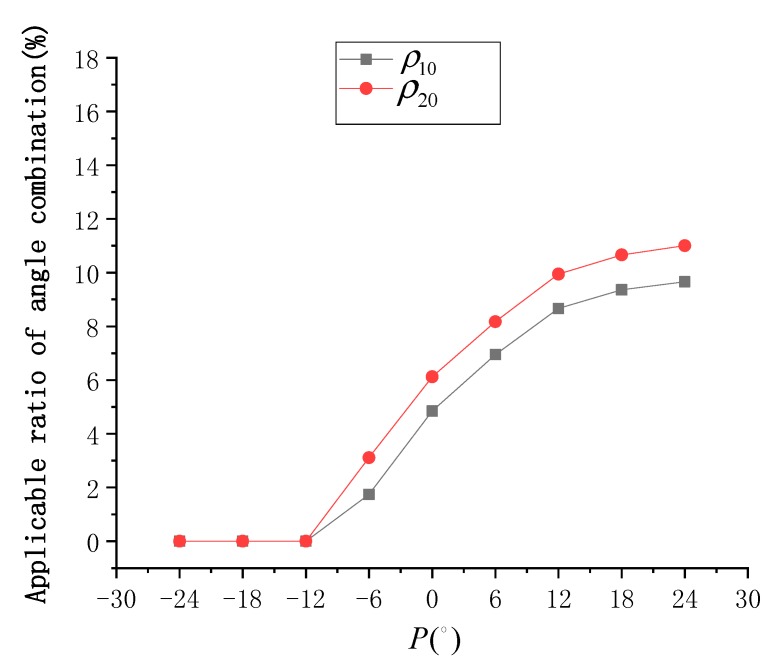
Statistical results of $\rho_{10}$ and $\rho_{20}$ at different pointing angles (*P*).

**Figure 6 sensors-19-02268-f006:**
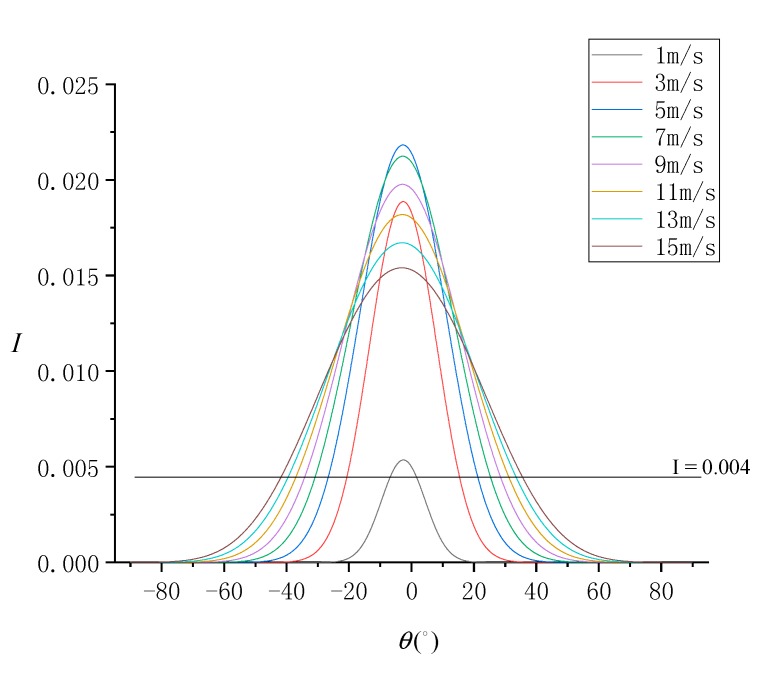
Normalized SG radiance (I) simulated under each SVA (θ) at different wind speeds.

**Figure 7 sensors-19-02268-f007:**
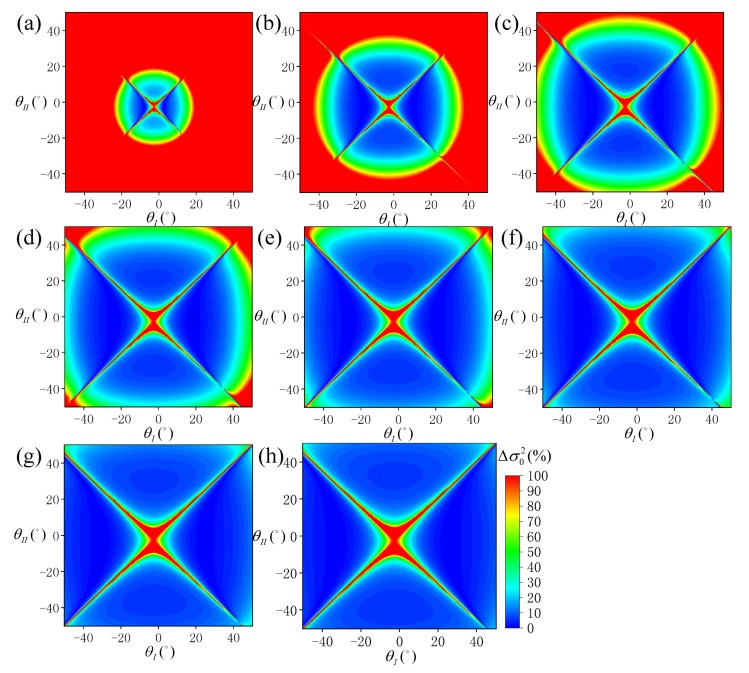
Distribution of SSR estimation errors ($\Delta \sigma_{0}^{2}$) simulated under each SVA combination ($\theta_{I}$,$\theta_{II}$) at different wind speeds: (**a**) *W* = 1 m/s, (**b**) *W* = 3 m/s, (**c**) *W* = 5 m/s, (**d**) *W* = 7 m/s, (**e**) *W* = 9 m/s, (**f**) *W* = 12 m/s, (**g**) *W* = 13 m/s, and (**h**) *W* = 15 m/s.

**Figure 8 sensors-19-02268-f008:**
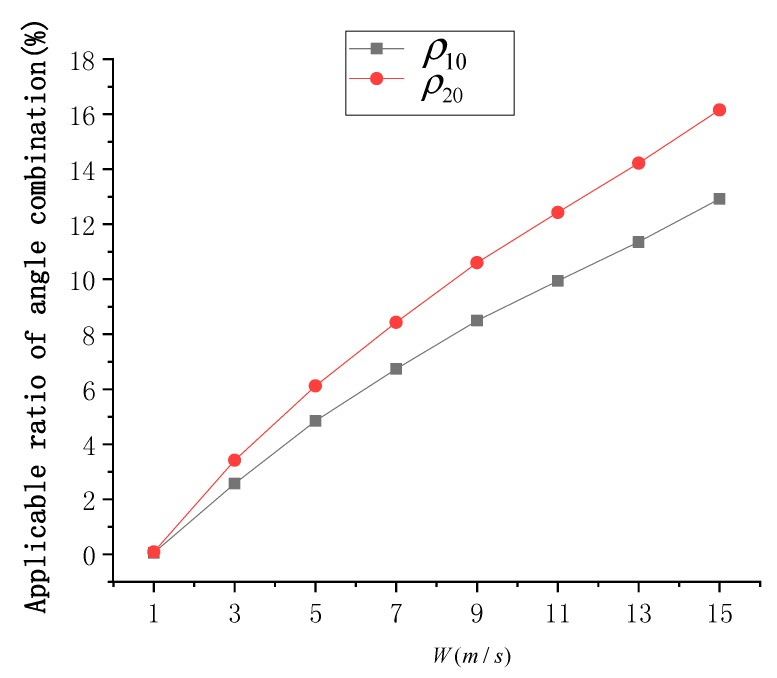
Statistical results of $\rho_{10}$ and $\rho_{20}$ at different wind speeds (*W*).

**Figure 9 sensors-19-02268-f009:**
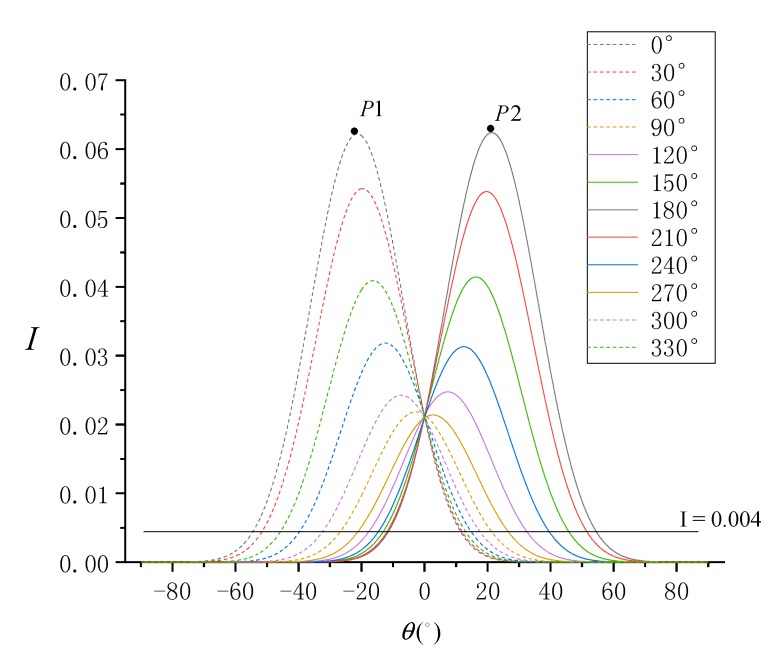
Normalized SG radiance (I) simulated by each SVA (θ) under different sun azimuths.

**Figure 10 sensors-19-02268-f010:**
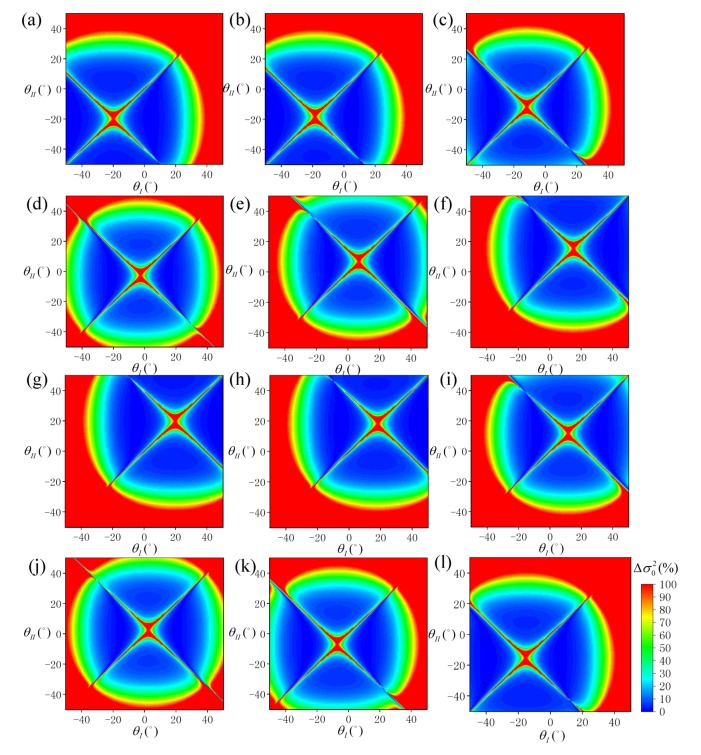
Distribution of SSR estimation error ($\Delta \sigma_{0}^{2}$) simulated under each SVA combination ($\theta_{I}$,$\theta_{II}$) at different sun azimuth angles: (**a**) $\;\phi_{0}$ = 0°, (**b**) $\phi_{0}$ = 30°, (**c**) $\phi_{0}$ = 60°, (**d**) $\phi_{0}$ = 90°, (**e**) $\phi_{0}$ = 120°, (**f**) $\phi_{0}$ = 150°, (**g**) $\phi_{0}$ = 180°, (**h**) $\phi_{0}$ = 210°, (**i**) $\phi_{0}$ = 240°, (**j**) $\phi_{0}$ = 270°, (**k**) $\phi_{0}$ = 300°, and (**l**) $\phi_{0}$ = 330°

**Figure 11 sensors-19-02268-f011:**
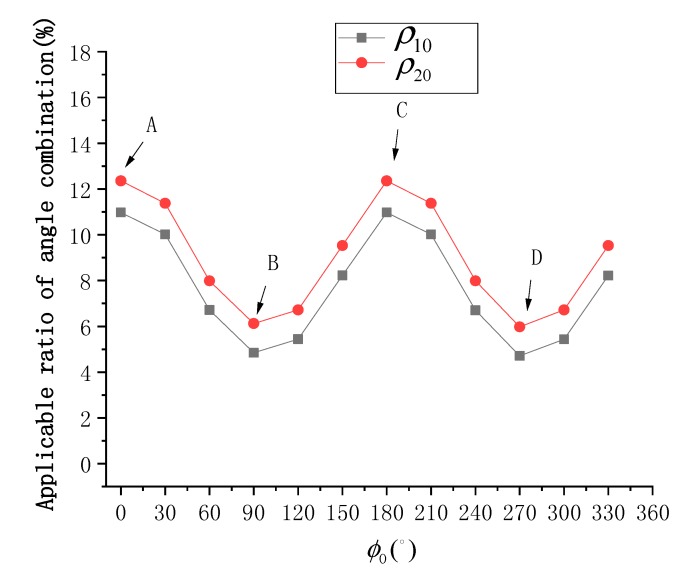
Statistical results of $\rho_{10}$ and $\rho_{20}$ at different sun azimuth angles ($\phi_{0}$).

**Figure 12 sensors-19-02268-f012:**
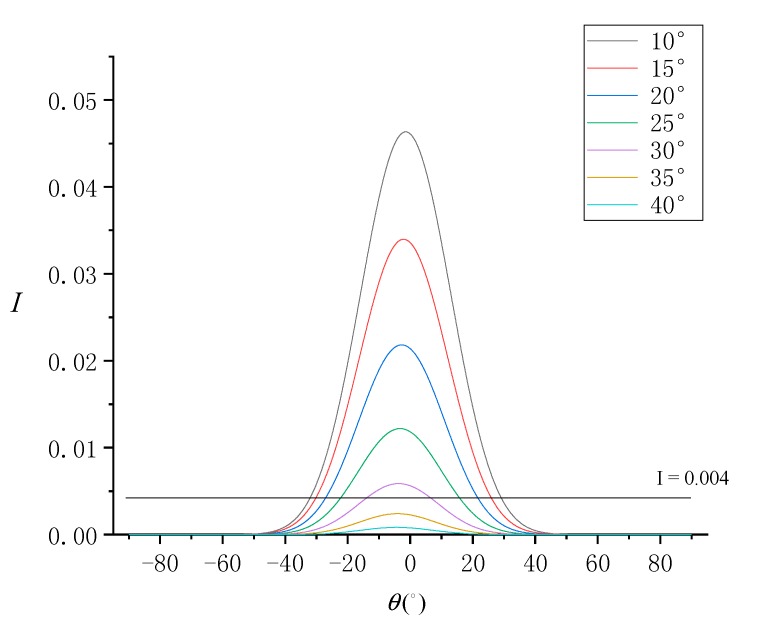
Normalized SG radiance (I) simulated under each SVA (θ) at different sun zenith angles.

**Figure 13 sensors-19-02268-f013:**
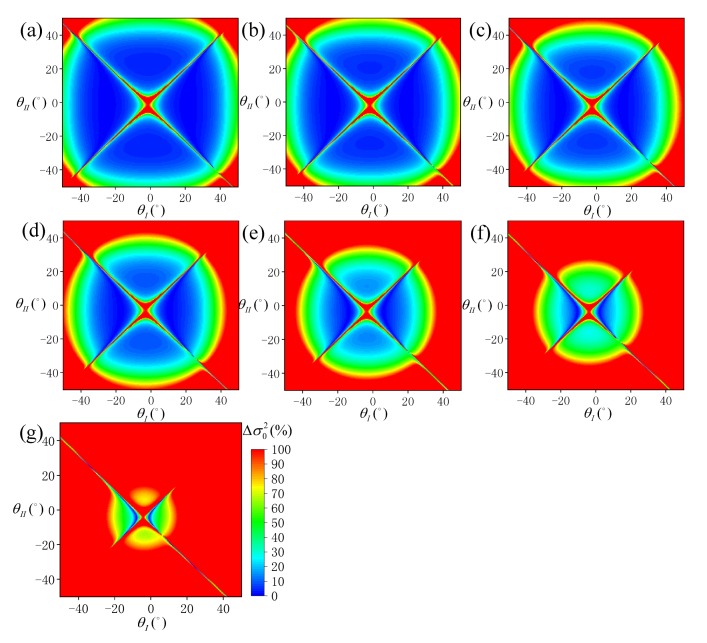
Distribution of SSR estimation error ($\Delta \sigma_{0}^{2}$) simulated under each SVA combination ($\theta_{I}$,$\theta_{II}$) at different sun zenith angles: (**a**) $\theta_{0}$ = 10°, (**b**) $\theta_{0}$ = 15°, (**c**) $\theta_{0}$ = 20°, (**d**) $\theta_{0}$ = 25°, (**e**) $\theta_{0}$ = 30°, (**f**) $\theta_{0}$ = 35° and (**g**) $\theta_{0}$ = 40°.

**Figure 14 sensors-19-02268-f014:**
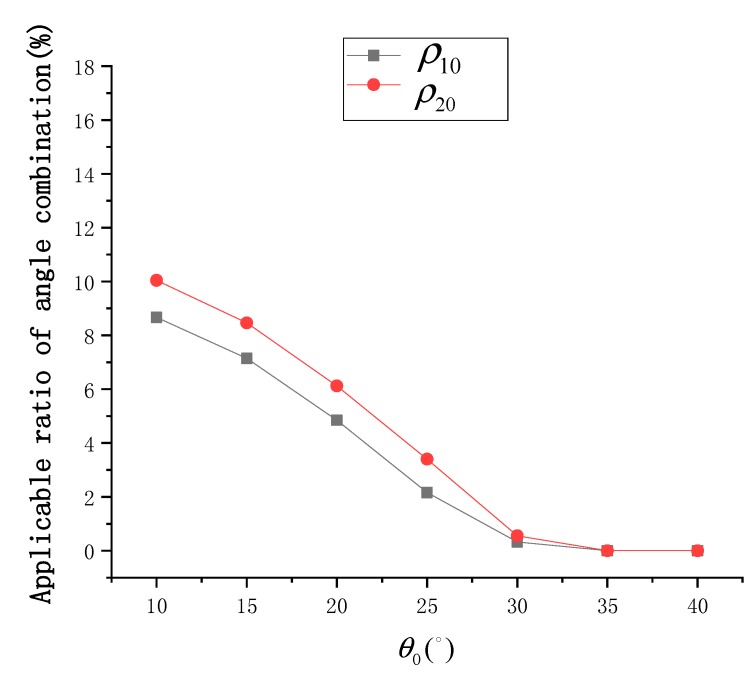
Statistical results of $\rho_{10}$ and $\rho_{20}$ at different sun zenith angles ($\theta_{0}$).

**Figure 15 sensors-19-02268-f015:**
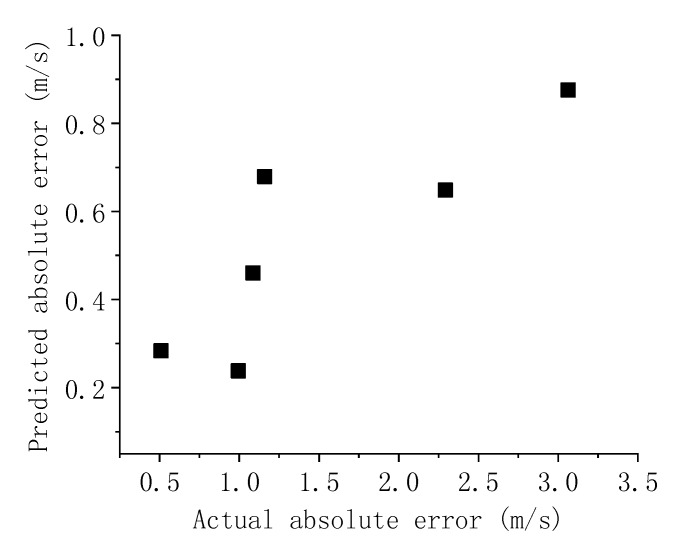
A comparison of the predicted and actual errors.

**Figure 16 sensors-19-02268-f016:**
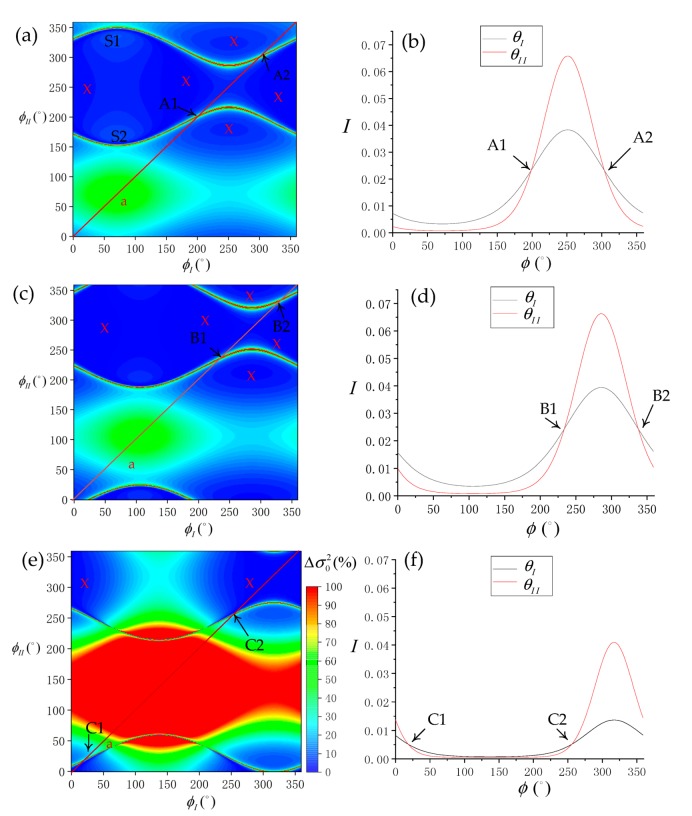
(**a**) Distribution of SSR estimation error ($\Delta \sigma_{0}^{2}$) at the simulation location [5° N, 120° E]. (**b**) The normalized SG radiance of the profile a in Figure 16a. (**c**) Distribution of SSR estimation error at simulation location [20° N, 120° E]. (**d**) The normalized SG radiance of the profile a in Figure 16c. (**e**) Distribution of SSR estimation error at simulation location [40° N, 120° E]. (**f**) The normalized SG radiance of the profile a in Figure 16f.

**Table 1 sensors-19-02268-t001:** Simulation model parameter settings.

Parameter	Value	Parameter Description
*IFOV*	$2.13\times 10^{-5}\;\left({}^{\circ}\right)$	Instantaneous field of view of ASTER
*S*	8 (°)	Scene orientation angle
*h*	$7.05\times 10^{5\;}\left(m\right)$	Satellite height
*m*	15 (m)	Image spatial resolution
*P*	0 (°)	Pointing angle
*W*	5 (m/s)	Sea surface wind speed
$\phi_{0}$	90 (°)	Sun azimuth angle
$\theta_{0}$	20 (°)	Sun zenith angle

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
