# Peer review of "On Optimal Imaging Angles in Multi-Angle Ocean Sun Glitter Remote-Sensing Platforms to Observe Sea Surface Roughness"

_sensors, 2019, doi:10.3390/s19102268_

Reviewer 1 Report

R1: [rows 194-208] provides numbered list of elements described in Figure 2. It would be correct to depict these numbers of elements in Fig.2.

R2: According to guidelines for authors and good practice, as well as to PRISMA checklist, the section "Materials and methods" must be provided. Reviewer recommendation is to provide this section as report describing experiment process (math. model construction, simulation process and used tools, measured errors, etc.). Many data are already depicted in sections 2-3, simply group them to see total picture of experiment (reviewer hasn't noticed applied tool for math. model simulation).

C1: [row 360-361] check font of text.

Author Response

First of all, thank you very much for your reminder. We have revised these mistakes. We apologize for these oversights.

Point 1:  [rows 194-208] provides numbered list of elements described in Figure 2. It would be correct to depict these numbers of elements in Fig.2.

Response 1: We have redrawn Fig.2,  and added the corresponding number to the figure.

Point 2: According to guidelines for authors and good practice, as well as to PRISMA checklist, the section "Materials and methods" must be provided. Reviewer recommendation is to provide this section as report describing experiment process (math. model construction, simulation process and used tools, measured errors, etc.). Many data are already depicted in sections 2-3, simply group them to see total picture of experiment (reviewer hasn't noticed applied tool for math. model simulation).

Response 2: We changed the title of section 2 from "Model" to "Materials and Methods" and described the simulation tools used in [rows 188-191]. So in this section, the math. model construction, simulation process and used tools are included.

Point 3:[row 360-361] check font of text.

Response 3:We checked out that there is an error and font error in these lines and it has been modified.

Reviewer 2 Report

This paper conducted simulations and a model developed to determine the optimal observation geometry for inversion from sun glitter images.  Then the model with optimized observation geometry was applied to field sun glitter images and was shown to give better inversion results.  The paper was well-written and was scientifically sound.  I think the paper is suitable for publication after a quick check through for errors in English.

Author Response

First of all, we thank you very much for your approval and encouragement for the revision of the previous version of our manuscript.

Previously, the paper had been modified by a professional English editing service. And I checked the full text for errors in English again and modified some errors.